# Non-Carcinogenic Risk Assessment for Heavy Metals in the Soil and Rice in the Vicinity of Dabaoshan Mine, South China

Huarong Zhao [1,2,*], Kangming Shi [1], Jianqiao Qin [3,*], Zikang Ren [1] and Guoliang Yang [1]

1   College of Environmental Science and Engineering, Guilin University of Technology, Guilin 541006, China; glutskm@163.com (K.S.); glutrzk@163.com (Z.R.); glutygl@163.com (G.Y.)
2   Collaborative Innovation Center for Water Pollution Control and Water Safety in Karst Area, Guilin University of Technology, Guilin 541006, China
3   College of Environmental and Chemical Engineering, Zhaoqing University, Zhaoqing 526061, China
*   Correspondence: zhaohuar@mail3.sysu.edu.cn (H.Z.); qinjianqiaosci@126.com (J.Q.)

**Abstract:** Heavy-metal pollution has attracted wide attention in recent years. The problem of heavy-metal pollution in the vicinity of the Dabaoshan mine, the largest polymetallic mine in South China, has attracted widespread attention. In this study, 38 samples of rice and paddy soil near the Dabaoshan mine were collected. The physical and chemical properties of the soil, including Cu, Cd, Zn, Pb, and Ni levels in the soil and rice, were analyzed. The heavy-metal baseline in paddy soil was analyzed by a normal Q–Q plot. The bioaccumulation factor of the rice was calculated. The non-carcinogenic risk of heavy metals was evaluated by calculating the hazard quotient (HQ). Threshold values of Cu, Cd, Zn, Pb, and Ni were 35.01, 0.51, 70.94, 59.78, and 16.34 mg/kg, respectively. The threshold values of Cu, Zn, and Pb were higher than the background value and lower than the secondary value of China's soil environmental quality standard. The threshold value of Cd was higher than both the background value and the secondary value of China's soil environmental quality standard. There was no significant threshold value for Ni in soil. The bioaccumulation factors of Cd, Zn, and Ni were straw > rice > husk. The bioaccumulation factors of Cu and Pb were straw > husk > rice. The HQ of Cd showed that the values for both adults and children were greater than 1, and the HQ for children was higher than that for adults. The HQs of Cu, Pb, Zn, and Ni were all less than 1. This indicated that Cu, Cd, Zn, and Pb pollution had occurred in the area, and that the Cd pollution was more serious. Therefore, it is necessary to strengthen land management, carry out the treatment of soil heavy metal pollution, and reduce the health risks of heavy metals in the study area.

**Keywords:** heavy metal; paddy soil; rice; hazard quotient; normal QQ plot; Dabaoshan mine

## 1. Introduction

Heavy metals are widely distributed in the world and may exist in the atmosphere, water, and sediments. The important sources of heavy metals in the environment are divided into natural sources and man-made sources. The natural sources are mainly from the weathering of rocks [1]. The man-made sources are metalliferous mining and smelting, agricultural materials, sewage sludge, metallurgical industries, fossil fuel burning, electronics, and chemical industries [2].

Mining is a main source of heavy-metal pollution, especially with rapid economic development. Tailings release heavy metals into the nearby soil, polluting soil and crops [3,4]. The atmosphere, water, and soil could be polluted by heavy metals during mining processes. Acid mine drainage, which is generated in mine excavation, could contaminate surface water and groundwater in the vicinity of a mine site. The soil vicinity of the mine will be polluted by heavy metals via dust sediment, acid mine drainage irrigation, and gangue dumps. Ultimately, heavy metals in environmental elements will accumulate in the soil.

Crops could be contaminated by heavy metals, if in the vicinity of the mine. Heavy metals in soil are transferred to crops as they grow [5]. In general, a complex mechanism is engaged when crops attempt to accumulate of heavy metals. Therefore, different types of crops can accumulate different heavy metals [6]. Rice is attested to be a highly accumulative crop for Cd [7–13]. Residents of affected areas will suffer from these heavy metals, as the metals are stably consumed when the rice is polluted. Moreover, excess heavy metal environmental pollutants (such as Cd and Pb) can harm animal health [14–16]. If the meat of these animals is consumed, it can also cause harm to human health.

The Dabaoshan mine is the largest open-pit polymetallic mine in southern China. Heavy metal (i.e., Cu, Cd, Zn, and Pb) contamination has occurred in the vicinity of the mine [17]. In previous studies, soil [18–21], water [22–24], and crops [25] have been found to be polluted. Residents have suffered from diseases caused by heavy metals [26–28]. Rice is the main food in this area. Rice consumption is a primary pathway for the heavy metals absorbed by residents. Risks from heavy metals consumed in rice should be a concern in this area. It is essential to research systematically both the pollution of heavy metals in rice and paddy soil and the residents harmed by heavy metals through consumption of the rice. The purposes of this work were (1) to evaluate the heavy-metal pollution levels of paddy soil, (2) to reveal heavy-metal enrichment in different parts of rice, and (3) to assess non-carcinogenic risks from heavy metals for residents through rice consumption.

## 2. Materials and Methods

### 2.1. Site Description

The study area is located in the vicinity of the Dabaoshan mine, Shaoguan City, Guangdong Province, South China (Figure 1). This area is located in a subtropical humid monsoon climatic zone. The annual rainfall and average temperature are 1762 mm and 20.3 °C, respectively. The Dabaoshan mine is the largest open-pit polymetallic mine in South China, and has been extensively mined for over 50 years. Acid mine drainage (AMD) of the Dabaoshan mine flows into the surface water, and the surface water has been used to irrigate the farmland for many years. Rice is the main product of the agricultural land and the stable food of the residents.

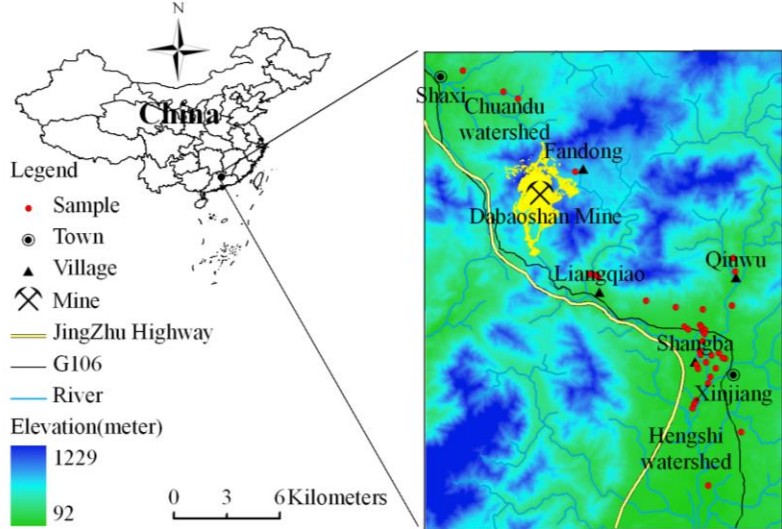

**Figure 1.** Study area and sample sites in the vicinity of Dabaoshan mine, South China.

### 2.2. Sampling

At harvest time, thirty-eight paddy soil samples were collected in the vicinity of the Dabaoshan mine. The depth of the paddy soil samples was 0–20 cm. Five sub-samples within 10 m$^2$ were mixed as a soil sample. The paddy soil, straw, and grain were collected

at each site and stored in polyethylene plastic bags. The locations were read by a portable GARMIN® global positioning system (GPS).

### 2.3. Chemical Analysis

The paddy soil samples were air-dried. Soil samples for pH analysis were screened with a 2 mm nylon screen. Part of the soil sample was ground with a mortar, passed through 100 screens, and stored in polyethylene plastic bags for analysis of heavy metals, organic matter, phosphorus, and nitrogen.

Straw and grain were washed with running water to remove the residuals first, and then washed with Milli-Q water three times. Straw and grain were dried at 105 °C for 1 h, then consistent weight at 60 °C, and the husk was separated from the rice. The samples of straw, husk, and rice were ground to tiny particles and stored in polyethylene plastic bags for heavy metals analysis.

Soil pH was measured by a Thermo pH meter (Thermo Orion 4 star, American), after mixing water and soil at a ratio of 1:2.5 [29]. The meter was calibrated by using reference buffer solution (Orion 910410 (pH = 4), Orion 910710 (pH = 7), and Orion 911010 (pH = 10)). Soil organic matter was determined by the Walkley–Black method [30]. The organic matter was oxidized to $CO_2$ with $K_2Cr_2O_7$ and $H_2SO_4$ at 180 °C. The unreacted $K_2Cr_2O_7$ was titrated with $FeCl_2$. The total nitrogen of the soil was determined by a nitrogen auto-analyzer (Kjeltec™-2300). The available phosphorus in the soil was extracted by $NaHCO_3$ and the phosphorus concentration was measured by the molybdenum-blue method [29].

Soil samples were digested by aqua regia. Straw, husk, and rice samples were digested by $HNO_3$-$HClO_4$ (4:1) [29]. Levels of Cu, Zn, and Ni in soil, straw, husk, and rice were measured by inductively coupled plasma optical emission spectrometry (ICP-OES). Levels of Cd and Pb were determined by graphite furnace atomic absorption spectroscopy (GFAAS).

Quality assurance included blank sample analysis, triplicate samples analysis, and certified reference material (GBW 10010 and GBW 08303) analysis. The certified reference materials were provided from the National Reference Materials Research Center. The recovery rates of heavy metals in standard substances were between 112% and 74%.

### 2.4. Method for Heavy Metal Baseline

Geochemical baselines provide means to distinguish pedogenetic and anthropogenic origins of heavy metals in environmental compartmentalization [31,32]. In general, methods to calculate the baseline can be classified as geochemical or statistical methods. A geochemical method requires expert knowledge about the investigated element under the prevailing environmental conditions. The statistical methods reference non-parametric methods, which include the Lepeltier method, relative cumulative frequency curves, normal range of a sample, regression technique, model analysis, 4σ-outlier tests, iterative 2σ techniques, and calculation of distribution functions [32].

The baseline for heavy metals in soil was based on a normal quantile–quantile (Q–Q) plot [33,34]. The obtained values were drawn on the x-axis, and the expected normal distribution was drawn on the y-axis. Curves usually have inflection points. The inflection points represent the critical value between natural-origin concentration and abnormal concentration [35].

### 2.5. Human Health Risk Calculation

The potential noncarcinogenic toxicity of heavy metal is estimated through its hazard quotient (HQ) [36]. HQ refers to the ratio of a chronic daily intake (CDI) to its reference dose (RfD) of heavy metals.

$$HQ = \frac{CDI}{RfD} \tag{1}$$

RfD is an index used to evaluate the risk of non-carcinogens; it is an estimate of the daily average exposure dose of exogenous chemicals in environmental media. Under a lifetime exposure to this dose level, the expected risk of non-carcinogenesis in life can

be reduced to an undetectable level. The RfD is adopted from the US EPA oral reference dose of heavy metal. The CDI is the sum of all the heavy metals ingested through the consumption of rice. The CDI can be calculated as:

$$CDI = \frac{EF \times ED \times FI_{rice} \times MC}{BW \times AT} \tag{2}$$

EF: exposure frequency (days/year); ED: average exposure duration (year); $FI_{rice}$: amount of rice ingestion (g/(person day)); MC: metal concentration in the rice (mg/kg); BW: average body weight (kg); and AT: average time (days).

Non-carcinogenic risk of heavy metals can be assessed using HQ. If the HQ is less than 1.0, there is no risk; but if the HQ is greater than 1.0, there is a risk [37].

### 2.6. Statistical Analysis

SPSS software (V16.0 for Windows) was used for data processing and statistical analysis.

### 3. Results and Discussion

#### 3.1. Heavy Metals in Paddy Soil

The characteristics and heavy-metal contents of the paddy soil are shown in Table 1. The pH of the paddy soil ranged from 3.67 to 6.48. The average content of total nitrogen and the available phosphorous of the paddy soil were 1374.37 (mg/kg) and 10.06 (mg/kg), respectively. The average organic matter of the paddy soils was 26.52 (g/kg), with a range from 48.55 (g/kg) to 10.41 (g/kg). The total concentrations (mg/kg) of Cu (216.14), Zn (199.03), Cd (0.52), Pb (184.09), and Ni (10.16) in the paddy soils ranged across 839.46–18.38, 478.11–35.99, 4.46–0.05, 1488.78–17.65, and 16.95–4.46, respectively. Zhou et al. [18] have studied soil characteristics of paddy soils in the vicinity of the village of Shangba. Average concentrations of Cu, Cd, Zn, Pb, pH, and organic matter were 56.7 (mg/kg), 2.68 (mg/kg), 1140 (mg/kg), 191 (mg/kg), 3.88, and 22.70 (g/kg), respectively. Except for Cu, the concentrations of other heavy metals were higher than those in this study. The results of organic matter and soil pH tests were similar to those in this study. Our results and previous studies showed that the content levels of heavy metals in paddy soils in this area is relatively high.

**Table 1.** Mean concentration, standard deviation, and range of Cu, Cd, Zn, Pb, Ni, total nitrogen, available phosphorous, organic matter, and pH in paddy soil in the vicinity of Dabaoshan mine.

|  | Mean ± SD | Range |  | Mean ± SD | Range |
|---|---|---|---|---|---|
| Cu (mg/kg) | 216.14 ± 208.33 | 839.46–18.38 | Total nitrogen (mg/kg) | 1374.37 ± 458.76 | 2445.45–532.44 |
| Zn (mg/kg) | 199.03 ± 131.59 | 478.11–35.99 | Available phosphorous (mg/kg) | 10.06 ± 12.51 | 59.47–0.07 |
| Cd (mg/kg) | 0.52 ± 0.78 | 4.46–0.05 | Organic matter (g/kg) | 26.52 ± 8.18 | 48.55–10.41 |
| Pb (mg/kg) | 184.09 ± 283.09 | 1488.78–17.65 | pH | 5.53 ± 0.66 | 6.48–3.67 |
| Ni (mg/kg) | 10.16 ± 3.09 | 16.95–4.46 |  |  |  |

Comparing the average heavy metals values to reference values is a good way to evaluate the level of environmental pollution. The Chinese soil environmental quality standards are used as the threshold values for soil heavy metal pollution. The threshold values are also shown in Figure 2. The threshold values of Cu, Cd, Zn, Pb, and Ni were 50 mg/kg, 200 mg/kg, 0.30 mg/kg, 250 mg/kg, and 40 mg/kg, respectively, which are adopted from the China Environmental Quality Standard for Soils (Grade II, pH < 6.5) (GB15618-1995). The average values of Cd and Cu were both higher than the threshold values. Moreover, the maximum values of Zn and Pb were higher than the threshold values. The maximum and average value of Ni were both below the threshold values. Guangdong Province is a low-soil-background-value area in China. The background values of Guangdong soil are lower than the national background values. The background values for soils in Guangdong Province are 14.7 mg/kg for Cu, 62.2 mg/kg for Zn, 0.046 mg/kg for Cd, 23.9 mg/kg for Pb, and 34.4 mg/kg for Ni (Figure 2) [31]. The average values

of heavy metal concentration in the paddy soils were above the background values of Guangdong Province, except for that of Ni. The results showed that the pollution degree of Cu and Cd in paddy soil was higher than that of Zn and Pb. However, there was no Ni pollution in the paddy soil.

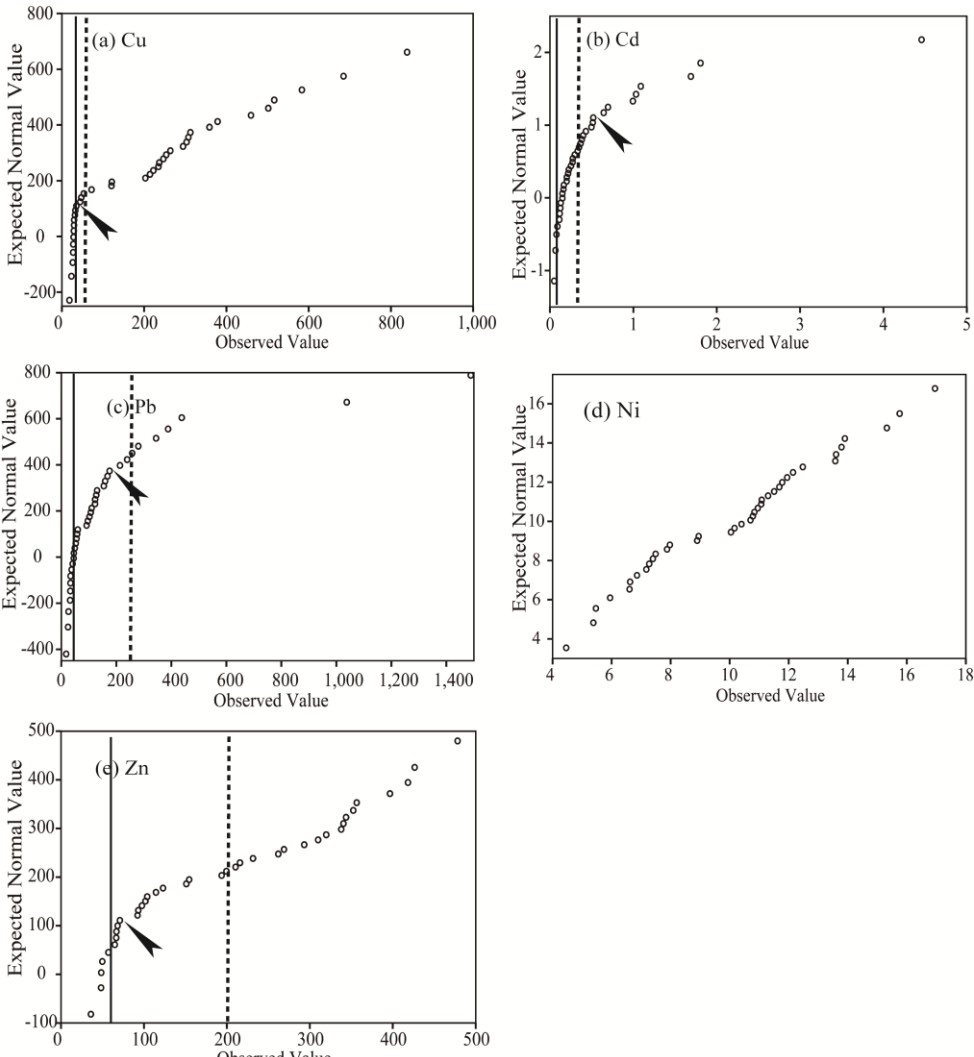

**Figure 2.** Normal Q–Q plots of heavy metals in paddy soil in the vicinity of Dabaoshan mine. The vertical solid line denotes the background levels, and the vertical dashed line denotes the Grade II China Environmental Quality Standard for Soils (GB 15618-1995) (pH < 6.5). The arrow indicates the threshold value.

In order to understand the relationship between soil properties and heavy metals, we analyzed the Pearson correlation between five heavy metals and soil properties (OM, pH, P, N) (Table 2). The results indicated that the correlativity among Cu, Cd, Zn, and Pb was significant in positive correlation. However, with Ni, there was no significant correlation with Cu, Cd, Zn, and Pb. In addition, pH exhibited a higher correlation with Cu, Cd, Zn, and Pb than did Ni in the paddy soil. Therefore, we believe that the Cu, Cd, Zn, and Pb come from same source. The correlation coefficient between OM and N was 0.960, which showed a positive significant correlation.

**Table 2.** Coefficient of Pearson correlation between heavy-metal levels and soil properties.

|     | Cu | Zn | Cd | Pb | Ni | P | N | OM | pH |
|-----|------|------|------|------|------|------|------|------|------|
| Cu | 1.000 | | | | | | | | |
| Zn | 0.756 ** | 1.000 | | | | | | | |
| Cd | 0.665 ** | 0.589 ** | 1.000 | | | | | | |
| Pb | 0.797 ** | 0.579 ** | 0.860 ** | 1.000 | | | | | |
| Ni | −0.075 | 0.270 | −0.143 | −0.163 | 1.000 | | | | |
| P | −0.305 | −0.351 * | −0.246 | −0.274 | −0.203 | 1.000 | | | |
| N | 0.178 | 0.123 | −0.088 | −0.173 | 0.285 | −0.378 * | 1.000 | | |
| OM | 0.226 | 0.111 | −0.087 | −0.144 | 0.162 | −0.354 * | 0.960 ** | 1.000 | |
| pH | −0.662 ** | −0.385 * | −0.465 ** | −0.562 ** | 0.167 | 0.332 * | −0.477 ** | −0.533 ** | 1.000 |

** Correlation is significant at the 0.01 level (two-tailed). * Correlation is significant at the 0.05 level (two-tailed).

Heavy metal contamination can be explained by calculating the background values of the paddy soil. The term "geochemical background" has no clear definition. However, it is generally agreed that geochemical background can provide basic guidance for monitoring soil environmental pollution [21]. The calculation of [mean ± 2 standard deviation] to estimate the background values was first used about 50 years after being proposed. The calculation is based on the assumption that the elements are in normal distribution and are in natural conditions in the soil. However, natural conditions and anthropogenic processes, the normal quantile–quantile (Q–Q) plot will provide a clearer answer to the geochemical background values [33].

In a normal Q–Q plot, the measured value is drawn on the X-axis, and the expected normal value is drawn on the Y-axis. These plots have powerful data visualizations, showing breakpoints and inflection points, suggesting possible different processes [34]. In addition, research showed that a normal Q–Q plot could avoid decreasing the impact of human activities on the heavy metals in the soil [31]. The upper limit of the collective background values is defined by the first bend of the slope on the plot curve [38]. The bend at the top of the figure (Figure 2) can be seen as the threshold value, and used to distinguish between natural samples (low concentration) and anthropogenic samples [21]. The 'breaks' on the plots of Cu, Zn, and Pb obviously exist, and their background values can be defined. However, the 'inflection point' on the Cd plot is too obscure to identify, and one therefore needs to refer to the national standard for soil and the Guangdong background values of Cd. Moreover, the curve trend of normal Q–Q plots of Ni is a line, and thus, one cannot find any 'inflection point' (Figure 2).

The soil geochemical background values of Cu, Cd, Zn, and Pb were determined from the first bend in the slope, and the value of Ni was determined by mean ± 2 standard deviations; the values were 35.01 mg/kg Cu, 70.94 mg/kg Zn, 0.51 mg/kg Cd, 59.78 mg/kg Pb, and 16.34 mg/kg Ni. The soil geochemical background values of the paddy soils are between the Chinese soil environmental quality standard for Grade II and the background values of soil in Guangdong Province, except for Cd and Ni. The maximum value of Ni is below both the Chinese soil environmental quality standard for Grade II and the background values of soils in Guangdong Province. The Cd value was above the Chinese soil environmental quality standard for Grade II and the soil background values of Guangdong Province. By studying the background values of heavy metals in the soil near the Dabaoshan mining area, it was found that the soil in this area was polluted by Cu, Cd, Zn, and Pb, among which Cd pollution was the most serious; Ni pollution was not obvious.

### 3.2. Heavy Metals in Rice

Heavy-metal presence in crops is not homogeneous. Heavy metals in leafy sections of plants typically are in much higher concentrations than in other parts [39]. The content of Cu in soil had less effect on levels in rice grains, while the contents of Cd and Cr had more effect on the rice grains. There are four steps to the transfer of heavy metals from soil to leaves: root absorption; root–ground transposition; redirection at nodes, and relocation at leaves. Plant genotype affects the transport of heavy metals from leaves to grains. At

the same Cd concentration in the soil, the Cd content in grains of Zheda 821 was always higher than that of Xiushui 817 [40]. The concentrations of heavy metals in rice straw, rice husk, and rice are shown in Table 3. The average concentrations of Zn, Cd, and Ni decrease in the order of straw > rice > husk, with the exception that average concentrations of Cu and Pb in the husk are higher than in rice. Heavy metal levels in straw were higher than in rice and husk, for straw consists of the stem and leafy sections. Simmons et al. [41] studied Cd and Zn in rice. Concentrations of Cd and Zn in unpolished rice grain were lower than in the stem. Zhuang et al. [42] studied heavy metals in rice harvested from the villages of Zhongxin, Fandong, Liangqiao, and Shangba in the vicinity of the Dabaoshan mine. Heavy metal levels in straw were higher than in husk and rice, while the levels of heavy metals in rice were lower than in straw and husk in most villages. The content levels of heavy metals in rice depend on whether the rice is polished or not. Heavy metal levels in unpolished rice are higher than in the polished.

**Table 3.** Heavy metal content of straw, husk, and rice (mg/kg).

| | Cu | | Zn | | Cd | | Pb | | Ni | |
|---|---|---|---|---|---|---|---|---|---|---|
| | Mean ± SD | Range | Mean ± SD | Range | Mean ± SD | Range | Mean ± SD | Range | Mean ± SD | Range |
| Straw | 31.27 ± 20.38 | 119.33–13.86 | 182.87 ± 228.99 | 1347.29–20.83 | 1.20 ± 1.42 | 5.05–0.06 | 4.16 ± 1.98 | 8.73–1.58 | 15.13 ± 21.53 | 123.35–2.09 |
| husk | 10.19 ± 1.76 | 15.13–7.13 | 19.45 ± 5.54 | 29.69–9.84 | 0.22 ± 0.24 | 0.98–0.02 | 0.96 ± 0.49 | 2.09–0.02 | 0.37 ± 0.51 | 2.14–0.01 |
| rice | 6.11 ± 1.79 | 10.53–3.41 | 24.48 ± 5.30 | 42.76–15.69 | 0.26 ± 0.35 | 1.40–0.01 | 0.14 ± 0.13 | 0.86–0.04 | 0.64 ± 0.59 | 2.14–0.02 |
| MAL * | 10 | | 50 | | 0.2 | | 0.2 | | 1 | |

* MAL: maximum allowable level of pollutants in food, as recommended by the China Ministry of Health (GB 2762-2005).

The highest concentrations of Cu, Pb, Cd, and Ni in the rice exceeded the maximum allowable levels for food pollutants in China (GB2762-2005). The results indicated that parts of the area were contaminated by Cu, Pb, Cd, and Ni. The mean concentration of Cd was greater than the maximum allowable level, which means Cd was the key pollutant in rice in the vicinity of the Dabaoshan mine. In order to reduce the harmful effects of heavy metals in the soil, heavy metal stabilizers can be added to the soil [43].

### 3.3. Bioaccumulation Factors from Paddy Soils to Rice

The effectiveness of heavy metal accumulation in plants can be evaluated using the bioaccumulation factor (BAF). BAF is defined as the ratio of the concentration of heavy metals in plants to that in soil. The heavy metal BAFs of straw, husk, and rice are shown in Figure 3. The heavy metal BAFs of straw, in decreasing order: Cd > Ni > Zn > Cu > Pb. The BAFs of Cd, Ni, and Zn in straws are above one. The BAFs of husk and rice, in decreasing order, are Cd > Zn > Cu > Ni > Pb. The BAFs of Cd in husk and rice are above one. The BAFs, in decreasing order, for different parts of the Oryza sativa L., are straw > rice > husk for Cd, Zn, and Ni, and straw > husk > rice for Cu and Pb.

The maximum value for BAF was Cd in straw, husk, and rice. The deviation of Cd was the highest in straw, husk, and rice. The results mean that the accumulation of Cd in rice is much higher than that of other heavy metals. Zhuang et al. [42] studied the BAFs of Cu, Cd, Zn, and Pb in crops of the study area; the BAF order was Cd > Zn > Cu > Pb, which agrees with the present study.

### 3.4. Human Health Risks of Rice Consume

HQ values were calculated by Equations (1) and (2). The RfD values of Cu, Cd, Zn, Pb, and Ni in Equation (1) were 0.04, 0.001, 0.3, 0.004, and 0.02 mg/kg day, respectively [44]. The average daily rice intake amounts for adults and children were 372 and 209.1 g/day, respectively; the average weights of adults and children were 65 and 32.7 kg, respectively; the exposure frequency was 365 days/year; the average contact times for adults and children were 365 × 70 and 365 × 12 day, respectively [25,26,45,46].

HQ values of Cu, Cd, Zn, Pb, and Ni for rice ingestion are shown in Figure 4. For adults and children, HQs of heavy metals from rice are, in decreasing order, Cd > Cu > Zn > Pb > Ni. The highest HQ value is associated with Cd. The HQ values of Cd for adults and

children are both above one. HQ values of Cu, Zn, Pb, and Ni are less than one for both adults and children. The HQs of Cu are approximately one for both adults and children. The results mean that Cd contamination occurred in rice in the study area. Residents who consume rice from the study area may suffer from diseases caused by Cd [47].

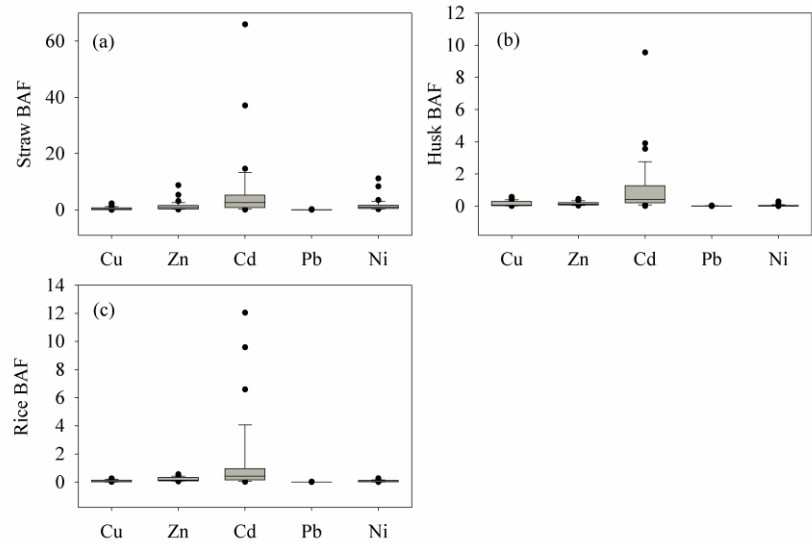

**Figure 3.** Box plots for the BAF of heavy metals in rice ((**a**) Straw, (**b**) Husk and (**c**) Rice) in the vicinity of Dabaoshan mine. The black points represent outliers.

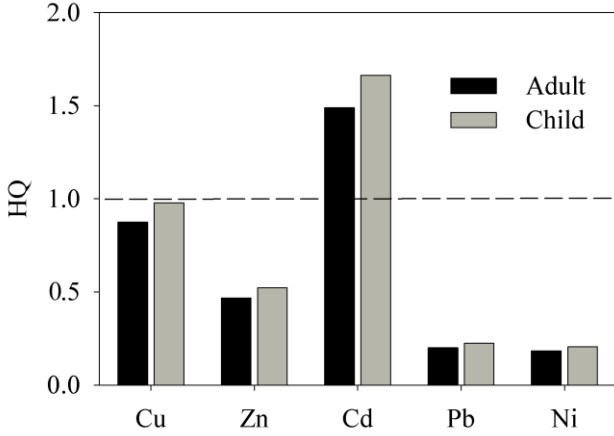

**Figure 4.** Human health risks of heavy metals for adults and children via rice consumption in the vicinity of Dabaoshan mine.

In the study area, the heavy metal most harmful to residents was Cd. Zhuang et al. [25,42] found that, in the study area, the HQ value of Cd in rice was the highest. Zhao et al. [26] have delineated spatial patterns of the HQs of Cu, Cd, Zn, and Pb in the study area, based on land use. Cd was the greatest health risk for agricultural land and residential land due to pollution from the Dabaoshan mine. The HQ of Cd was one order of magnitude larger than that of Cu, Zn, Pb, and Ni. The human health risk of Cd was spatially simulated according to soil pH and soil organic carbon in the study area [46]. The higher risk area overlapped with lower soil pH and higher soil-organic-carbon areas. These studies indicated that Cd was the main pollutant causing human health risks in the study area.

HQ values for children were greater than those for adults, when considered for rice consumption. The results indicated that children were at higher risk of non-carcinogenic effects than were adults [42]. The HQs of children were higher than those for adults, when

considered for food consumption in previous studies [48,49], indicating that children are more sensitive to heavy metal risks than are adults.

## 4. Conclusions

Paddy soils and rice were contaminated by heavy metals in the vicinity of the Dabaoshan mine in southern China. Threshold values of Cu, Zn, and Pb in the paddy soil exceeded the natural background levels for Guangdong Province. The threshold value of Cd in the paddy soil exceeded the Grade II China Environmental Quality Standard for Soils (GB 15618-1995) (pH < 6.5). Higher BAF for Cd led to increased Cd content in rice compared to other metals tested. The Cd content in rice exceeded the maximum allowable level for the China food standard. The HQ value of Cd is greater than 1 for children and adults who eat rice grown near the Dabaoshan mine area. The HQ values of Cd for children are greater than those for adults. The primary pollutant in the study area was Cd. Cd pollution poses a potential risk to the residents. Therefore, it is necessary to strengthen soil management and carry out soil remediation to reduce the non-carcinogenic risks of Cd pollution, especially for children.

**Author Contributions:** Project administration, writing—review and editing, H.Z.; conceptualization, J.Q.; data analysis and review, K.S.; methodology and investigation, Z.R. and G.Y. All authors have read and agreed to the published version of the manuscript.

**Funding:** This work was funded by the Guangxi Key R&D Program (Guike-AB22080093, Guike-AB22035075, and Guike-AB21075007) and the Guilin Key R&D Program (20210212-2).

**Data Availability Statement:** The data presented in this study are available on request from the corresponding author.

**Acknowledgments:** The authors gratefully acknowledge the assistance of the Guangxi Science and Technology Department and the Guilin Science and Technology Bureau.

**Conflicts of Interest:** The authors declare no conflict of interest.

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
