# Peer review of "Non-Carcinogenic Risk Assessment for Heavy Metals in the Soil and Rice in the Vicinity of Dabaoshan Mine, South China"

_processes, doi:10.3390/pr11071970_

Round 1

Reviewer 1 Report

In this manuscript, the concentrations of Cu, Zn, Cd, Pb and Ni in paddy soil and rice in Dabaoshan mining area and their possible health risks were studied. The results indicated that the pollution of Cu, Zn, Cd and Zn were occurred and the primary pollutant was Cd in the study area and Cd pollution has potential serious harm to the health of residents in the study area. The selection of sampling points, sampling methods and statistical analysis are basically reasonable, but the analysis of results needs to be further refined to provide the basis for assumptions, guesses, and relevant literature supporting the interpretation of possible causes. The following issues also need to be explained and improved.

1. The abstract fails to adequately highlight the key points of the manuscript, and only briefly touches upon the experimental results. It is recommended that the content of the abstract be enriched.

2. To enhance the search ability, it is suggested to incorporate HQ values and Normal Q-Q plot into the keywords.

3. In Introduction part, the language should be simplified and the writing level improved.

4. In 76 line, pay close attention to the written content and please thoroughly review the manuscript.

5. In 219-220 lines, the reasons for the accumulation of heavy metals in various locations can be further elucidated, and through more comprehensive literatures, the research depth of this manuscript can be enhanced.

6. In 254-281 lines, In the study of human toxicity, it is necessary to investigate the content of various heavy metal forms. The degree of harm caused by different valence states of heavy metals varies, which enhances the rigor of experimental design. Suggest using BCR method for heavy metal speciation analysis.

7. In 296-297 lines, the manuscript please give some specific advices, do not generalize.

Author Response

1.The abstract fails to adequately highlight the key points of the manuscript, and only briefly touches upon the experimental results. It is recommended that the content of the abstract be enriched.

We have rewritten the abstract.

  1. To enhance the search ability, it is suggested to incorporate HQ values and Normal Q-Q plot into the keywords.

HQ and normal Q-Q plot have been added to keywords.

  1. In Introduction part, the language should be simplified and the writing level improved.

We have revised the introduction part.

  1. In 76 line, pay close attention to the written content and please thoroughly review the manuscript.

We have revised the sentence.

  1. In 219-220 lines, the reasons for the accumulation of heavy metals in various locations can be further elucidated, and through more comprehensive literatures, the research depth of this manuscript can be enhanced.

The reasons for the accumulation of heavy metals in various locations have been added. “There are four steps in the transfer of heavy metals from soil to leaves: root absorption; root to shoot translocation; redirection at nodes and remobilization from leaves. However, plant genotype affects the transport of heavy metals from leaves to grains”.

  1. In 254-281 lines, In the study of human toxicity, it is necessary to investigate the content of various heavy metal forms. The degree of harm caused by different valence states of heavy metals varies, which enhances the rigor of experimental design. Suggest using BCR method for heavy metal speciation analysis.

The suggestions made by the reviewers are very good, and we will consider BCR method for heavy metal risk research in future research.

  1. In 296-297 lines, the manuscript please give some specific advices, do not generalize.

We have rewritten this section to recommend enhanced soil management and soil remediation to reduce the risk of heavy metal contamination.

Reviewer 2 Report

Manuscript ID: processes-2411800

Journal: Processes 

Manuscript Title: Heavy metals in paddy soil and rice and their health risks vicinity to

Dabaoshan Mine, Southern China

This study is interesting and meaningful. But before the manuscript is considered for acceptance, it needs major revision, especially in results and discussion section. In addition, pay particular attention to tenses, single, and plural. My comments are as follows.

Section Abstract

1. L 14: Revise “Cd, Pb and Ni” to “Cd, Pb, and Ni”.

2. L 14-15: Revise “and their possible” to in order to explore their possible”. Are the changes right?

3. L 19-20: “Reasonable relationships”?

4. L 25: Please rewrite “there is no obvious that threshold value of Ni was observed.”.

5. L 29: Zn appears twice in the following contents: “Cu, Zn, Cd and Zn”.

6. L 29: Revise “were occurred” to “occurred”.

Section Introduction

7. L 34: Revise “the world. Heavy metals may” to “the world and may”.

8. L 47: Revise be transfer to be transferred.

9. L 55: Revise “caused by heavy metal” to “caused by heavy metals. Is the change right?

10. L 61: Revise “to reveal heavy metal accumulated” to “to reveal heavy metal accumulation.

11. It is recommended to supplement that excess heavy metal environmental pollutants (such as Cd and Pb) can harm animal health (Liu et al., 2023; Cui et al., 2023; Yu et al., 2022). in introduction section to further support research background of the manuscript.

Yuhao Liu et al. Cadmium exposure caused cardiotoxicity in common carps (Cyprinus carpio L.): miR-9-5p, oxidative stress, energetic impairment, mitochondrial division/fusion imbalance, inflammation, and autophagy.  Fish & Shellfish Immunology, 2023, 108853.  doi: 10.1016/j.fsi.2023.108853.

Jiawen Cui et al. Cadmium induced time-dependent kidney injury in common carp via mitochondrial pathway: impaired mitochondrial energy metabolism and mitochondrion-dependent apoptosis. Aquatic Toxicology, 2023, 106570. doi: 10.1016/j.aquatox.2023.106570.

Meijin Yu et al. HSP27-HSP40-HSP70-HSP90 pathway participated in molecular mechanism of selenium alleviating lead-caused oxidative damage and proteotoxicity in chicken bursa of Fabricius,. Animal Biotechnology, 2022, https://doi.org/10.1080/10495398.2022.2155175.

Section Materials and methods

12. L 67: Revise “20.3℃” to “20.3 °C.

13. L 76: Revise “in 10 m2” to “in 10 m2. Is the change right?

14. L 80: Revise “a 2mm” to “a 2 mm”.

15. L 84: Revise “washed by running water” to “washed with running water”.

16. L 106: Regarding “The recoveries of heavy metals were 112-74%”, “112%”?

17. L 111: Revise “method required expert” to “method requires expert”.

Section Results and discussion

18. L 143: Revise “paddy soil are” to “paddy soil were”.

19. L 144: Revise “paddy soils is” to “paddy soils was”. Please pay attention to tense.

20. L 147: Revise “1488.78-17.65 and 16.95-4.46.” to “1488.78-17.65, and 16.95-4.46, respectively.”. Are the changes right?

21. L 159: Revise “mg/kg and 40mg/k” to “mg/kg, and 40 mg/k”.

22. L 173: Revise “The result showed” to “The results showed”.

23. L 178: Revise “achieve to” to “achieved to”.

24. L 182: Revise “The heavy metal contaminations” to Heavy metal contamination”.

25. L 194: Revise “research shows” to “research showed.

26. L 205: Revise “heavy metal” to “heavy metals.

27. L 210: Revise “the values are” to “the values were”.

28. L 140-217: Regarding “3.1 Heavy metals in paddy soil” section, the authors do not discuss the authors' research results well, but only reviews other people's researches and showed their own results, without discussing what conclusions the authors' research results draw. It is suggested that the authors reorganize this part of the discussion. There is a similar problem later in the discussion.

29. L 224: Revise “the straw consisted of” to “Straw consists of ”.

30. L 236: Revise “area contaminated” to “area were contaminated”.

31. L 255-256: Regarding “HQ values were calculated in accordance with equations 1-2. The RfD of Cu, Zn, Cd, Pb and Ni were 0.04, 0.3, 0.001, 0.004 and 0.02 mg/kg day, respectively [36]”, the authors cited a reference. I wonder whether the calculation results in the above sentence are the calculation results of this manuscript or the results of the reference. There are also similar doubts in other part in discussion section. .

32. L 257: Revise “and 209.1 g/day;” to “and 209.1 g/day, respectively;”. Are the changes right? Please pay attention to similar problems.

33. L 261: Revise “ingestion are shown” to “ingestion were shown”. Pay attention to tense

34. L 265: Revise “The result means” to “The results meant.

35. L 266-267: Please supplement reference to support “Residents consume rice may suffer from diseases caused by Cd in the study area.”.

36. In the discussion, it is suggested that the authors more clearly express which information is your own research results and conclusions, and which information is the research results and conclusions of other scholars

37. L 277: Please rewrite “HQ values of children were higher than that of adults”. Please note the consistency of single and plural.

38. L 278: Please supplement reference to support “the risk of non-carcinogenic effects”.

39. L 280-281: Regarding The results indicate that children were more sensitive to risks of heavy metals.”, are The results the results of the authors' research or those of other scholars?

40. L 283: Revise “of heavy metal for adult” to “of heavy metals for adults.

Section Concusion

41. L 287: Revise “Cu, Zn and Pb” to “Cu, Zn, and Pb”.

42. L 289: Revise “pH<6.5” to “pH < 6.5”.

Moderate editing of English language required

Author Response

Section Abstract

According to the opinions of other reviewers, we rewrote the abstract.

1.L 14: Revise “Cd, Pb and Ni” to “Cd, Pb, and Ni”.

We have revised it.

  1. L 14-15: Revise “and their possible” to “in order to explore their possible”. Are the changes right?

We have revised it.

  1. L 19-20: “Reasonable relationships”?

We have revised it

  1. L 25: Please rewrite “there is no obvious that threshold value of Ni was observed.”.

We have revised it

  1. L 29: “Zn” appears twice in the following contents: “Cu, Zn, Cd and Zn”.

We have revised it.

  1. L 29: Revise “were occurred” to “occurred”.

We have revised it.

Section Introduction

  1. L 34: Revise “the world. Heavy metals may” to “the world and may”.

We have revised it.

  1. L 47: Revise “be transfer” to “be transferred”.

We have revised it.

  1. L 55: Revise “caused by heavy metal” to “caused by heavy metals”. Is the change right?

We have revised it.

  1. L 61: Revise “to reveal heavy metal accumulated” to “to reveal heavy metal accumulation”.

. We have revised it.

  1. It is recommended to supplement that “excess heavy metal environmental pollutants (such as Cd and Pb) can harm animal health (Liu et al., 2023; Cui et al., 2023; Yu et al., 2022).” in introduction section to further support research background of the manuscript.

Yuhao Liu et al. Cadmium exposure caused cardiotoxicity in common carps (Cyprinus carpio L.): miR-9-5p, oxidative stress, energetic impairment, mitochondrial division/fusion imbalance, inflammation, and autophagy.  Fish & Shellfish Immunology, 2023, 108853.  doi: 10.1016/j.fsi.2023.108853.

Jiawen Cui et al. Cadmium induced time-dependent kidney injury in common carp via mitochondrial pathway: impaired mitochondrial energy metabolism and mitochondrion-dependent apoptosis. Aquatic Toxicology, 2023, 106570. doi: 10.1016/j.aquatox.2023.106570.

Meijin Yu et al. HSP27-HSP40-HSP70-HSP90 pathway participated in molecular mechanism of selenium alleviating lead-caused oxidative damage and proteotoxicity in chicken bursa of Fabricius,. Animal Biotechnology, 2022, https://doi.org/10.1080/10495398.2022.2155175.

We have added the effects of heavy metals on animal health to the manuscript。

Section Materials and methods

  1. L 67: Revise “20.3℃” to “20.3 °C”.

We have revised it.

  1. L 76: Revise “in 10 m2” to “in 10 m2”. Is the change right?

We have revised it.

  1. L 80: Revise “a 2mm” to “a 2 mm”.

We have revised it.

  1. L 84: Revise “washed by running water” to “washed with running water”.

We have revised it.

  1. L 106: Regarding “The recoveries of heavy metals were 112-74%”, “112%”?

We have revised it as “The recovery rate of heavy metals was between 112% and 74%.”

  1. L 111: Revise “method required expert” to “method requires expert”.

We have revised it.

Section Results and discussion

  1. L 143: Revise “paddy soil are” to “paddy soil were”.

We have revised it.

  1. L 144: Revise “paddy soils is” to “paddy soils was”. Please pay attention to tense.

We have revised it.

  1. L 147: Revise “1488.78-17.65 and 16.95-4.46.” to “1488.78-17.65, and 16.95-4.46, respectively.”. Are the changes right?

We have revised it.

  1. L 159: Revise “mg/kg and 40mg/k” to “mg/kg, and 40 mg/k”.

We have revised it.

  1. L 173: Revise “The result showed” to “The results showed”.

We have revised it.

  1. L 178: Revise “achieve to” to “achieved to”.

We have revised it.

  1. L 182: Revise “The heavy metal contaminations” to “Heavy metal contamination”.

We have revised it.

  1. L 194: Revise “research shows” to “research showed”.

We have revised it.

  1. L 205: Revise “heavy metal” to “heavy metals”.

We have revised it.

  1. L 210: Revise “the values are” to “the values were”.

We have revised it.

  1. L 140-217: Regarding “3.1 Heavy metals in paddy soil” section, the authors do not discuss the authors' research results well, but only reviews other people's researches and showed their own results, without discussing what conclusions the authors' research results draw. It is suggested that the authors reorganize this part of the discussion. There is a similar problem later in the discussion.

We have modified this section and marked the changes in red.

  1. L 224: Revise “the straw consisted of” to “Straw consists of”.

We have revised it.

  1. L 236: Revise “area contaminated” to “area were contaminated”.

We have revised it.

  1. L 255-256: Regarding “HQ values were calculated in accordance with equations 1-2. The RfD of Cu, Zn, Cd, Pb and Ni were 0.04, 0.3, 0.001, 0.004 and 0.02 mg/kg day, respectively [36]”, the authors cited a reference. I wonder whether the calculation results in the above sentence are the calculation results of this manuscript or the results of the reference. There are also similar doubts in other part in discussion section.

RfD is the abbreviation of reference dose. We have revised the sentence as “The RfD values of Cu, Zn, Cd, Pb and Ni in equation 1 were 0.04, 0.3, 0.001, 0.004, and 0.02 mg/kg day, respectively.”

  1. L 257: Revise “and 209.1 g/day;” to “and 209.1 g/day, respectively;”. Are the changes right? Please pay attention to similar problems.

We have revised it.

  1. L 261: Revise “ingestion are shown” to “ingestion were shown”. Pay attention to tense

We have revised it.

  1. L 265: Revise “The result means” to “The results meant”.

We have revised it.

  1. L 266-267: Please supplement reference to support “Residents consume rice may suffer from diseases caused by Cd in the study area.”.

We have added references.

  1. In the discussion, it is suggested that the authors more clearly express which information is your own research results and conclusions, and which information is the research results and conclusions of other scholars

We have revised the discussion part.

  1. L 277: Please rewrite “HQ values of children were higher than that of adults”. Please note the consistency of single and plural.

We have rewritten the sentence as “HQ value of children was higher than that of adults”

  1. L 278: Please supplement reference to support “the risk of non-carcinogenic effects”.

We have added references.

  1. L 280-281: Regarding “The results indicate that children were more sensitive to risks of heavy metals.”, are “The results” the results of the authors' research or those of other scholars?

We have revised the sentence as “HQs of children were higher than adults through food consumption in previous studies, indicate that children were more sensitive to risks of heavy metals.”

  1. L 283: Revise “of heavy metal for adult” to “of heavy metals for adults”.

We have revised it.

Section Concusion

  1. L 287: Revise “Cu, Zn and Pb” to “Cu, Zn, and Pb”.

We have revised it.

  1. L 289: Revise “pH<6.5” to “pH < 6.5”.

We have revised it.

Reviewer 3 Report

1. The authors can consult the following research papers to make the introduction more extensive between lines 46-51.

i) https://doi.org/10.1016/j.jes.2022.10.038

ii) https://doi.org/10.1016/j.chemosphere.2023.138267

2. In the Materials and methods section the authors should mention the depth of soil sampling.

3. The authors should mention about the QA/QC details of analysis of heavy metals in form of Table in supplementary may be.

4. For HQ calculation a citation of reference is required. The authors might consult https://doi.org/10.1016/j.chemosphere.2019.06.088

5.  Rather than considering 1 for HQ the authors might consider it 0.5 as the source of contamination will not be from consumption of rice only. Other food materials and drinking water might be a source not considered here. The authors can consult the following research papers

https://doi.org/10.1016/j.chemosphere.2019.06.088

https://doi.org/10.1080/15320383.2019.1661353

6. other citations:

https://doi.org/10.1016/j.ecoenv.2014.01.001

https://doi.org/10.1080/15226514.2017.1413328

https://doi.org/10.1016/j.apsoil.2010.03.006

https://doi.org/10.1016/j.apgeochem.2013.09.001

https://pubs.acs.org/doi/abs/10.1021/acs.jafc.8b01525

No comments

Author Response

  1. The authors can consult the following research papers to make the introduction more extensive between lines 46-51.
  2. i) https://doi.org/10.1016/j.jes.2022.10.038
  3. ii) https://doi.org/10.1016/j.chemosphere.2023.138267

The content of the above literature has been added to the introduction.

  1. In the Materials and methods section the authors should mention the depth of soil sampling.

We have added the description of sampling depth.

  1. The authors should mention about the QA/QC details of analysis of heavy metals in form of Table in supplementary may be.

The accuracy of heavy metal analysis is ensured by blank sample analysis, parallel sample analysis and reference material analysis.

  1. For HQ calculation a citation of reference is required. The authors might consult https://doi.org/10.1016/j.chemosphere.2019.06.088

In HQ calculation, we have referred to the above literature.

  1. Rather than considering 1 for HQ the authors might consider it 0.5 as the source of contamination will not be from consumption of rice only. Other food materials and drinking water might be a source not considered here. The authors can consult the following research papers

https://doi.org/10.1016/j.chemosphere.2019.06.088

https://doi.org/10.1080/15320383.2019.1661353

We have referred to the above literatures

  1. other citations:

https://doi.org/10.1016/j.ecoenv.2014.01.001

https://doi.org/10.1080/15226514.2017.1413328

https://doi.org/10.1016/j.apsoil.2010.03.006

https://doi.org/10.1016/j.apgeochem.2013.09.001

https://pubs.acs.org/doi/abs/10.1021/acs.jafc.8b01525

The manuscript refers to some of the above literature.

Reviewer 4 Report

Processes-2411800

Upon review of the manuscript by Zhao et al., I found the quality of English to be very poor and needs extensive editing by a professional before the paper can be reviewed for scientific merit and consideration for publication in the journal Processes.

Processes-2411800

Upon review of the manuscript by Zhao et al., I found the quality of English to be very poor and needs extensive editing by a professional before the paper can be reviewed for scientific merit and consideration for publication in the journal Processes.

Author Response

Upon review of the manuscript by Zhao et al., I found the quality of English to be very poor and needs extensive editing by a professional before the paper can be reviewed for scientific merit and consideration for publication in the journal Processes.

 We checked and revised the language of the manuscript.

Round 2

Reviewer 1 Report

The writing and readability of the second draft have been improved. Most of the suggestions I have made have been revised, but no more experiments have been added. If no experiments are added, the article will also be complete.

Further polishing in English shoud do to increase readability.

Author Response

Further polishing in English shoud do to increase readability.

We have polished the manuscript

Reviewer 2 Report

Manuscript ID: processes-2411800v2

Manuscript Title: Heavy metals in paddy soil and rice and their health risks vicinity to Dabaoshan Mine, Southern China

Journal: Processes

I suggest that the editorial department accepted the manuscript after the authors correcting some minor errors.

1. L 15-16: Revise “The heavy metal baseline” to Heavy metal baseline”.

2. L 17: Revise “The health risk” to Health risk”.

3. L 19: Revise “are higher” to were higher”.

4. L 21: Revise “is higher” to was higher”.

5. L 22: Revise “There is no” to “There was no”.

6. L 26: Revise “are all” to were all”.

7. L 27: Revise “It indicates” to “It indicated.

8. L 27: Revise “pollution have occurred” to “pollution occurred”.

9. L 28: Revise “is more” to was more”.

10. The references cited by the authors in the text are marked with serial numbers, but the authors does not provide the serial number of each reference in the reference section. It is difficult for the reader to confirm the exact information of the references cited in the text.

11. L 83: Revise “Part of the soil sample was ground” to “Part of soil samples were ground”.

12. L 85: Revise “phosphorus and” to “phosphorus, and”.

13. L 119: Revise “The observed values were plotted” to “The obtained values were plotted”.

14. L 154: Revise “Our results and previous studies show” to “Our results and previous studies showed.

15. L 165: Revise “are both higher” to were both higher”.

16. L 166: Revise are higher than to were higher than.

17. L 167: Revise are all below to were all below.

18. L 201: Please provide special information of the figure.

19. L 222-224: Revise “it is found that the soil in this area has been polluted by copper, zinc, cadmium and lead, among which cadmium pollution is the most serious and nickel pollution is not obvious” to “it was found that the soil in this area was polluted by Cu, Zn, Cd, and Pb, among which Cd pollution was the most serious and Ni pollution was not obvious.

20. L 227: Revise “The content of copper” to “The content of Cu.

21. L 228: Revise “the content of cadmium and chromium” to “the content of Cd and Cr.

22. L 229: Revise “There are” to “There were.

23. L 232: Revise “cadmium concentration in the soil, the cadmium content” to Cd concentration in the soil, the Cd content”.

24. L 234: Revise “are shown in” to were shown in.

25. L 284: Revise “was cadmium” to was Cd. 

26. L 285: Revise “cadmium in rice is” to Cd in rice was”.

Minor editing of English language required

Author Response

I suggest that the editorial department accepted the manuscript after the authors correcting some minor errors.

  1. L 15-16: Revise “The heavy metal baseline” to “Heavy metal baseline”.

We have revised it.

  1. L 17: Revise “The health risk” to “Health risk”.

We have revised it.

  1. L 19: Revise “are higher” to “were higher”.

We have revised it.

  1. L 21: Revise “is higher” to “was higher”.

We have revised it.

  1. L 22: Revise “There is no” to “There was no”.

We have revised it.

  1. L 26: Revise “are all” to “were all”.

We have revised it.

  1. L 27: Revise “It indicates” to “It indicated”.

We have revised it.

  1. L 27: Revise “pollution have occurred” to “pollution occurred”.

We have revised it.

  1. L 28: Revise “is more” to “was more”.

We have revised it.

  1. The references cited by the authors in the text are marked with serial numbers, but the authors does not provide the serial number of each reference in the reference section. It is difficult for the reader to confirm the exact information of the references cited in the text.

The references of the manuscript are inserted by reference management software, and each reference has been provided with a serial number.

  1. L 83: Revise “Part of the soil sample was ground” to “Part of soil samples were ground”.

We have revised it.

  1. L 85: Revise “phosphorus and” to “phosphorus, and”.

We have revised it.

  1. L 119: Revise “The observed values were plotted” to “The obtained values were plotted”.

We have revised it.

  1. L 154: Revise “Our results and previous studies show” to “Our results and previous studies showed”.

We have revised it.

  1. L 165: Revise “are both higher” to “were both higher”.

We have revised it.

  1. L 166: Revise “are higher than” to “were higher than”.

We have revised it.

  1. L 167: Revise “are all below” to “were all below”.

We have revised it.

  1. L 201: Please provide special information of “the figure”.

We have provided the special information of “the figure”.

  1. L 222-224: Revise “it is found that the soil in this area has been polluted by copper, zinc, cadmium and lead, among which cadmium pollution is the most serious and nickel pollution is not obvious” to “it was found that the soil in this area was polluted by Cu, Zn, Cd, and Pb, among which Cd pollution was the most serious and Ni pollution was not obvious”.

We have revised it.

  1. L 227: Revise “The content of copper” to “The content of Cu”.

We have revised it.

  1. L 228: Revise “the content of cadmium and chromium” to “the content of Cd and Cr”.

We have revised it.

  1. L 229: Revise “There are” to “There were”.

We have revised it.

  1. L 232: Revise “cadmium concentration in the soil, the cadmium content” to “Cd concentration in the soil, the Cd content”.

We have revised it.

  1. L 234: Revise “are shown in” to “were shown in”.

We have revised it.

  1. L 284: Revise “was cadmium” to “was Cd”.

We have revised it.

  1. L 285: Revise “cadmium in rice is” to “Cd in rice was”.

We have revised it.

Reviewer 4 Report

processes-2411800: The paper by Zhao et al. quantified multiple heavy metals in the soil and rice grains grown in the vicinity of the Dabaoshan Mine, Southern China and estimated human health risk using standard HQ method. Below are my major comments that need to be addressed before the paper can be considered for publication in Processes.

1) Refine the title to improve clarity. One suggestion would be "Non-carcinogenic risk assessment of heavy metals in the soil and rice in the vicinity of Dabaoshan Mine, Southern China".

2) Abstract: Reflect "non-carcinogenic risk" in the abstract. Define HQ as non-carcinogenic risk.

3) Abstract line 22. The authors said "there is no significant threshold value for Ni". Did they mean threshold value for Ni in soil and/or rice/food, or did they mean reference value (RfD)? Please clarify this in Abstract. The USEPA many have published values for Ni that I suggest to explore and revise the paper accordingly. Here is the RfD for Ni: https://www.atsdr.cdc.gov/ToxProfiles/tp15-c8.pdf

4) I suggest tabulating all the threshold values and RfDs for all the analyzed metals and name it Table 2. Move current Table 2 to Supporting information.

5) Line 128. Define RfD (add more text for clarity).

6) More discussion is needed for Eq. 1. What values were used for EF, ED, BW and FI and provide justification for using those values? Clarify this in Methods section.

7) Line 306: Amend the statement "Cd could be enriched in rice, so that the Cd content of rice was relatively high" as "Higher BAF for Cd led to increased Cd content in rice compared to other metals tested."

8) Line 310: Amend "HQ values of Cd of children..." as "HQ values of Cd for children..."

9) Line 287. "The highest risk occurred in agriculture and residential land which was using Cd. " Provide more information on the source of Cd in ag. and residential lands.

10) For risk assessment, it is critical to assess uncertainty of the risk values calculated. This can be done in many ways; one of which could be using multiple regression model followed by bootstrapping as was accomplished in this paper: https://www.sciencedirect.com/science/article/abs/pii/S0048969721019641?via%3Dihub. The uncertainty must be addressed before the paper can be considered for publication.

There is still room for improvement.

Author Response

1) Refine the title to improve clarity. One suggestion would be "Non-carcinogenic risk assessment of heavy metals in the soil and rice in the vicinity of Dabaoshan Mine, Southern China".

We have changed the title of the manuscript to the title suggested by the reviewers

2) Abstract: Reflect "non-carcinogenic risk" in the abstract. Define HQ as non-carcinogenic risk.

We have revised line 12 of the manuscript to define HQ as a non-carcinogenic risk

3) Abstract line 22. The authors said "there is no significant threshold value for Ni". Did they mean threshold value for Ni in soil and/or rice/food, or did they mean reference value (RfD)? Please clarify this in Abstract. The USEPA many have published values for Ni that I suggest to explore and revise the paper accordingly. Here is the RfD for Ni: https://www.atsdr.cdc.gov/ToxProfiles/tp15-c8.pdf

There is no obvious threshold value for Ni in the soil, and we have modified 23 rows.

4) I suggest tabulating all the threshold values and RfDs for all the analyzed metals and name it Table 2. Move current Table 2 to Supporting information.

The threshold is the value used to describe the content of heavy metals in the soil, while the reference dose is the parameter used to calculate the health risk, which belong to different sections. It is not appropriate to put both in one table.

5) Line 128. Define RfD (add more text for clarity).

The manuscript has added an explanation of the RfD.

6) More discussion is needed for Eq. 1. What values were used for EF, ED, BW and FI and provide justification for using those values? Clarify this in Methods section.

This section mainly introduces the calculation method, and the value of each parameter has been explained in 3.4.

7) Line 306: Amend the statement "Cd could be enriched in rice, so that the Cd content of rice was relatively high" as "Higher BAF for Cd led to increased Cd content in rice compared to other metals tested."

We have revised it.

8) Line 310: Amend "HQ values of Cd of children..." as "HQ values of Cd for children..."

We have revised it.

9) Line 287. "The highest risk occurred in agriculture and residential land which was using Cd. " Provide more information on the source of Cd in ag. and residential lands.

We have revised the sentence.

10) For risk assessment, it is critical to assess uncertainty of the risk values calculated. This can be done in many ways; one of which could be using multiple regression model followed by bootstrapping as was accomplished in this paper: https://www.sciencedirect.com/science/article/abs/pii/S0048969721019641?via%3Dihub. The uncertainty must be addressed before the paper can be considered for publication.

It is true that there is uncertainty in risk assessment, but this paper does not study the parameters of heavy metal risk calculation. Instead, it is calculated by using the existing parameters.